# Copy Number Variations as Determinants of Colorectal Tumor Progression in Liquid Biopsies

**DOI:** 10.3390/ijms24021738

**Published:** 2023-01-16

**Authors:** Jessica Debattista, Laura Grech, Christian Scerri, Godfrey Grech

**Affiliations:** 1Department of Pathology, Faculty of Medicine and Surgery, University of Malta, MSD 2080 Msida, Malta; 2Department of Physiology and Biochemistry, Faculty of Medicine and Surgery, University of Malta, MSD 2080 Msida, Malta

**Keywords:** CRC, CNV, liquid biopsies, exosomes, ctDNA, CTC, long-RNA

## Abstract

Over the years, increasing evidence has shown that copy number variations (CNVs) play an important role in the pathogenesis and prognosis of Colorectal Cancer (CRC). Colorectal adenomas are highly prevalent lesions, but only 5% of these adenomas ever progress to carcinoma. This review summarizes the different CNVs associated with adenoma-carcinoma CRC progression and with CRC staging. Characterization of CNVs in circulating free-RNA and in blood-derived exosomes augers well with the potential of using such assays for patient management and early detection of metastasis. To overcome the limitations related to tissue biopsies and tumor heterogeneity, using CNVs to characterize tumor-derived materials in biofluids provides less invasive sampling methods and a sample that collectively represents multiple tumor sites in heterogeneous samples. Liquid biopsies provide a source of circulating tumor DNA (ctDNA), circulating tumor cells (CTCs), tumor-derived exosomes (TDE), circulating free RNA, and non-coding RNA. This review provides an overview of the current diagnostic and predictive models from liquid biopsies.

## 1. Introduction

Colorectal cancer (CRC) is the second most deadly cancer, with over 1.8 million newly diagnosed cases per year and over 900,000 deaths worldwide in 2020 [1]. CRC is a disorder that occurs exclusively in the colon or rectum and is caused by the colon’s aberrant proliferation of glandular cells. These rapidly developing cells give rise to a benign adenoma which, through several distinct pathways, can advance to cancer and metastasize [2,3]. Once the adenocarcinoma becomes invasive, it can spread to other parts via the blood and lymphatic arteries. The liver is the most common organ of distant metastasis, followed by the peritoneum cavity and the lungs [4].

## 2. CNVs in CRC Disease Progression

### 2.1. CNVs in Adenoma-Carcinoma

Over the years, increasing evidence has shown that copy number variations (CNVs), also known as copy number alterations (CNAs), play an important role in the pathogenesis and prognosis of CRC [5,6]. CNVs are structural variants in DNA sequences consisting mainly of duplications, deletions and insertions. In CRC, a number of CNVs are known to play an important role in the development of adenocarcinomas, with some CNVs more noticeable in early carcinogenesis while others are more prominent during disease progression and metastasis [6].

In the early 1990s, Vogelstein proposed that several sequential genetic mutations in key signaling pathways are responsible for the neoplastic alterations of the normal colonic epithelial cells leading to adenoma formation [2]. One of the earliest genetic alterations in the development of CRC adenoma is the loss of function in the *APC* gene (5q), which disrupts the WNT-signaling pathway. More than 80% of sporadic CRC have somatic APC mutations, while a loss of 5q is reported in about 30–40% of CRC cases [7,8]. Mutant APC gives rise to the accumulation of ß-catenin protein in the cytoplasm. The accumulated ß-catenin protein is translocated to the cell nucleus, which complexes with TCF/LEF inducing overactivation of the WNT downstream effectors [9].

Colorectal adenomas are highly prevalent lesions, but only 5% of these adenomas ever progress to carcinoma. Cross-sectional studies on several colorectal tumors consisting of non-progressed colorectal adenomas, progressed adenomas, and colorectal carcinomas showed that a number of CNVs were strongly associated with adenoma to carcinoma progression. These included gains in 7p (*EGFR*), 8q (*MYC),* 13q (*CDX2* and *PDX1*), and 20q and losses of 1p, 8p, 15q, 17p (*TP53*) and 18q (*DCC, SMAD2* and *SMAD4*) [C] [10,11,12]. In addition, 4q, 14q, and 20p were specifically found when comparing adenomas to stage 1 CRC [13]. The most common gains/amplifications and losses implemented in the adenoma to carcinoma progression summarized in Table 1.

Of interest, in a study on 297 adenomas, 23 to 36% of advanced adenomas had two or more CNVs when compared to only 1.7 to 4.8% in non-advanced adenomas [14]. In fact, out of 1699 CRC tumors, it was found that in almost half (47%) of the CRCs, loss of n8p co-occurred with a gain in 8q. The co-occurrence of these two abnormalities has significantly elevated the odds for the formation of carcinomas [15]. Gains of 8q leads to increased expression of *C-MYC*, potentiating tumorigenesis, while loss of 8p results in decreased expression of *DLC1*, a tumor suppressor gene [16,17]. In Stage II and Stage III CRC tumors, a loss of 18q heterozygosity (LOH) is commonly observed and is associated with a poor prognosis [3,18,19,20,21]. Several tumor suppressor genes (TSG) are present on the 18q loci and, therefore, the loss of 18q plays a significant role in CRC pathogenesis [13,22], exemplified by the *SMAD* genes which encode downstream signal transducers involved in TGF-ß their alterations may confer resistance to TGF-β and contribute to tumorigenesis [12].

Gains in 20q are found in more than 91% of CRC tumors [13] and are involved in transforming adenoma to carcinoma indicating poor prognosis [23]. Gains in 20q are accompanied by overexpression of a number of genes, mainly *C20orf24*, *AURKA*, *TH1L*, *ADRM1*, *C20orf20, TCFL5,* and *TPX2* [24,25]. *AURKA* (Aurora kinase A on 20q13.2) is a serine/threonine kinase family member involved in mitotic entry, bipolar spindle formation, centrosome maturation control and segregation during mitosis [26] and, therefore, its overexpression, has an impact on chromosomal segregation and cell growth [27]. It was found that the overexpression of *AURKA* together with *TPX2* promotes the transformation from adenoma to carcinoma [25]. Most of the amplified genes in 20q belong to various signaling pathways that may be involved in the CRC progression [27]. For example, *AURKA* is involved in the Wnt and Ras-MAPK pathways and the *TH1L*, is involved in the MEK/ERK pathway [28,29]. 

### 2.2. CNVs in Carcinoma-mCRC

Deletions and losses of 3p, 5p, and 4p are associated with Stage III carcinoma, exemplified by *FHIT* (3p14.2), which is found in 24.3% of stage III as compared to stage II tumors (3.3%) [30]. This suggests that deletions of *FHIT* play an important role in CRC progression. In addition, deletions of 5p were associated with advanced T, N or M stages [31].

Metastasis involves several distinct steps whereby the tumor cells travel through the circulation to implant and grow at secondary tissue sites. These steps involve local invasion, epithelial–mesenchymal transition (EMT), cell migration, intravasation into the lymphatic or hematogenous system, the survival of the tumor cell in the vascular system, the extravasation from the vasculature to distal tissue and colonization to secondary organs [32,33]. At the time of diagnosis, 20% of CRC patients already have metastases, with the liver being the most common site of metastasis (70%), followed by the thorax (32%), peritoneum (21%) and, in some instances, metastasis can also occur to the lungs, nervous system, skin and soft tissue [34]. 

A number of studies have been carried out to investigate the differences in CNVs between the primary CRC tumor and the metastatic site. Although a large number of CNVs are present in both the primary tumor and the metastatic tumor, a number of CNVs are only found in the metastatic tumor (Table 2). Comparison between primary and liver metastases in 20 patients showed that gains in 13q, 20q, 8q, 20p, 7p, 7q, and 1q and losses in 8p, 18q, 18p, 1p, 17p, and 4p were present in both primary and corresponding liver metastatic tumor. Copy number gains in Xq and 6p and losses in 14q and 22q were only seen in metastatic liver tumors [35]. In 2018, Kawamata [36] performed a genome-wide chromosomal copy number assessment between primary CRC tumors and paired liver metastases from 16 patients. The copy number status of 65.4% (123/188) genes was shared between the primary and the metastasis. These included *APC*, *TP53*, *KRAS*, *EGFR*, *VEGFA*, and *SMAD4*. A number of amplifications and losses were observed only in the metastatic tumor and not in the primary tumor; these included amplifications of *TGFBR2*, *CTNNB1* and *FHIT* on chromosome 3p, amplification of *PIK3CA*, *CBLB*, and *KALRN* on 3q, amplification of *FGFR*1 on 8p, amplifications of *CDK8* on chromosome 13q, and amplification of *ERBB2*, which encodes HER2 on 17q. Loss of *RBFOX1* on 16p was also noted. When comparing primary tumors to their metastatic liver tumor, it was found that an increased copy number of chromosomes 6p and 17q was associated with liver metastasis [37].

Comparisons between primary tumors and metastatic liver tumors in 27 metastatic CRC patients showed that a number of chromosomal alterations are shared between the two. These include gains of chromosomes 7, 8q, 13q, and 20q and losses of the 1p, 8p, 14q, and 18q. In contrast, del 22q and del 17p were more frequently found in patients with liver metastasis than in their matched primary tumor [38]. A meta-analysis of 373 primary tumors and 102 liver metastases was performed to identify chromosomal variants that differentiate among the Duke’s stages of CRC and those responsible for the progression into liver metastasis. It was suggested that losses at 17p and 18p and gains of 8q, 13q, and 20q occur early in establishing primary CRCs, whereas loss of 4p is associated with the transition from Dukes’ A to B–D. Deletion of 8p and gains of 7p and 17q are correlated with the transition from the primary tumor to liver metastasis, whereas losses of 14q and gains of 1q, 11, 12p, and 19 are late events [39]. Gains of 1q were accompanied by an increase in *TGFB2* amplification, while gains of chromosome 11 were accompanied by amplification of *MCAM*. A detailed analysis of the primary and metastatic tumors showed a high level of discordance between the two. *CARD11* (7p.22.2) and *MMP9* (20q.13.12) CNVs were found to be diploid in the primary tumor but increased in metastatic sites—mainly the liver and the retroperitoneum—while *SMAD4* (18q21.2) was decreased specifically in the metastasis [40]. Comparison between primary tumor and lymph node metastasis shows that patients with lymph node metastasis had a significantly higher chromosomal gain in the 8q23-23 locus [41]. When comparing lung metastases with their corresponding primary tumor, it was shown that deletions at 3p, 8p, 12q, 17q, and 21q21 and gains at 5p were observed more often in the lung metastasis [42].

**Table 1 ijms-24-01738-t001:** The most common chromosomal alterations and associated genes involved in adenoma to carcinoma progression.

	CNVs	Associated Gene	Reference
Gain	7p	*EGFR*	[11]
Gain	8q	*c-MYC*	[10,16,17]
		*LYN*	[13]
Gain	13q	*CDX2*	[10,16,17]
		*POLR1D*	[13]
		*PDX1*	[11]
Gain	20q	*AURKA*	[13,14]
		*TH1L*	[14]
		*ADRM1*	[14]
		*C20orf20*	[14]
		*TCFL5*	[14]
		*TPX2*	[25]
		*PMPEA1*	[13]
		*MMP9*	[13]
		*MYBL2*	[13,39]
		*UBE2C*	[13]
Loss	1p		[11]
Loss	3p	*FHIT*	[30]
Loss	4p		[30]
Loss	5p		[31]
Loss	8p	*CSMD*	[13]
		*DLC1*	[10,16]
Loss	14q		[13]
Loss	15q		[10]
Loss	17p	*TP53*	[10]
Loss	18q	*DCC*	[12]
		*SMAD2*	[12]
		*SMAD4*	[12]
		*CCDC68*	[13]
		*SERPINB7*	[39]
		*CTDP1*	[39]
Loss	20p		[13]

**Table 2 ijms-24-01738-t002:** The most common chromosomal alterations and their associated genes are involved in carcinoma to metastatic progression. Liver and lung tend to be the most common metastatic sites and are also characterized by different gains and losses.

Copy Number Alterations	Associated Gene	Metastatic Site	Reference
Gain of 1q	*TGFB2*	Liver metastasis	[39]
Gain of 3p	*TGFBR2* *CTNNB1* *FHIT*	Liver metastasis Liver metastasis Liver metastasis	[36]
Gain 3q	*CBLB* *KALRN* *PIK3CA*	Liver metastasis Liver metastasis Liver metastasis	[36]
Gain 5p		Lung metastasis	[36]
Gain 6p		Liver metastasis	[13,35]
Gain 8p	*FGFR1*	Liver metastasis	[36]
Gain 11p		Liver metastasis	[39]
Gain 11q	*MCAM*	Liver metastasis	[39]
Gain 12p		Liver metastasis	[39]
Gain 13q	*CDK8*	Liver metastasis	[36]
Gain 17q	*ERBB2*	Liver metastasis	[36]
Gain Xq			[35]
Loss 14q		Liver metastasis	[35,39]
Loss 16p	*RBFOX1*	Liver metastasis	[36]
Loss 17p		Liver metastasis	[38]
Loss 22q		Liver metastasis	[35,38]

CNVs, apart from being involved in the pathogenesis of CRC initiation and progression, also have a prognostic and predictive value. Therefore, it is of utmost importance that one determines the CNVs present in solid tumors. Although multiple advances have been made in methods used for detecting CNVs, one has to keep in mind that there are some technical limitations related to the methods used and how one interprets the results [5]. Manual microdissection of the tumors sample is based on the histological assessment of tissue biopsies and tumor resections by pathologists. Such samples are obtained via invasive procedures, mainly through surgical resections, which remained a mainstay in the cure and control of solid tumors [43]. However, despite their usefulness in the clinical scenario, surgical resections do not fully satisfy the criteria to monitor the recurrence of the disease. Apart from being invasive, surgeries play a significant role in tumor growth and metastasis, as proposed by the ‘seed and soil’ hypothesis of Stephen Paget [44]. Additionally, surgical resections are not always accessible/available, and limited information is available regarding tumors heterogeneity. It has been well demonstrated that, over time, tumors acquire different genetic and epigenetic profiles that may differ from the original tumors. Tissue-based genetic testing on a single resection specimen may not always reflect the tumors’ biology. Therefore, the absence of CNVs is either a false negative result related to sampling issues and tumor heterogeneity or a true positive with no CNVs. These limitations might lead to mistreatment and misdiagnosis, especially with the rapidly evolving targeted therapeutics. 

One way to overcome the limitations related to tissue biopsies and tumor heterogeneity is to look at the CNVs in detail, identify genes of interest, and profile them for expression using a circulating source of tumor-derived materials in biofluids also less invasive than the conventional techniques. These types of liquid biopsy include mRNA, circulating tumor DNA (ctDNA), circulating tumor cells (CTCs), tumor-derived exosomes, circulating free RNA, non-coding RNA and tumor-educated platelets (Figure 1).

## 3. Diagnostic and Predictive Models in Liquid Biopsies

### 3.1. mRNA Diagnostic Models in CRC

Circulating biochemical molecules, such as mRNA in blood, can be used as a biomarker in solid tumors. The expression of *EGFR* transcripts in the blood of 16 patients with CRC was assessed. All patients who expressed *EGFR* transcripts in their peripheral blood were found to express the EGFR protein in their primary tumor. EGFR transcripts in the primary tumor were also confirmed by using RT-PCR [45]. Xu et al. [20] evaluated the transcript level of *carcinoembryonic antigen (CEA)*, *cytokeratin 19 (CK19)* and *cytokeratin 20 (CK20)* in the peripheral blood of 168 CRC patients and 30 healthy controls. *CEA* expression was detected in 35.8% of CRC patients and 3.3% of healthy controls, *CK19* was detected in 41.9% of CRC patients and also in 3.3% of the healthy controls, while *CK20* was detected in 28.3% of the CRC patients and 6.7% of the controls. *CEA* and *CK20* mRNA increased with advancing Dukes stages. A study on blood samples from 370 CRC patients and 350 controls was carried out to explore the clinical significance of the expression of *CEA* mRNA. More than half of the CRC patients (53.8%) were positive for *CEA* mRNA. *CEA* mRNA also correlated with tumor staging and lymph node metastasis. Higher *CEA* mRNA expression was seen in patients with TNM stage III and IV compared to TNM stage I and II representing 61% of patients with lymph node metastasis [46].

A study was carried out to evaluate the expression levels of a multiple mRNA marker panel in the peripheral blood of 27 CRC patients. These included *CK19, CK20, CEA, REG4*, *uPA*, and *TIAM1*. All genes were expressed in more than 70% of the CRC patients with *CEA, CK19*, and *REG4* expressed in 77.8%, *TIAM1*, and *uPA* expressed in 74%, while *CK20* expressed in 70.4% of the patients. The overexpression of either *CK19, CEA* or *CK20* as a single marker correlated with lymph node metastasis, while overexpression of more than three mRNA markers was correlated with TNM stage [47]. 

In addition to the above markers, expression levels of 28 cancer-associated genes were investigated in the peripheral blood of 111 CRC patients and 227 non-cancer controls. Out of these 28 genes, five genes—*MDM2*, *DUSP6*, *CPEB4*, *MMD* and *E1F2S3*—were significantly associated with CRC [48]. Microarray analysis on mRNA from peripheral blood of healthy controls and CRC patients, showed that several genes were upregulated in CRC patients and not in controls. These included genes involved in cell adhesion, like *CD44, TGFβ, ICAM1*, and genes involved in cell proliferation, such as *IFITM1, IFITM2, TIMP1.* In addition, cells involved in intracellular signal transduction, such as *S100A11, filamin A* and *DDEF1*. UGDH involved in metabolism and *SLC26A2* involved in sulfate transport were both downregulated [49]. 

### 3.2. ctDNA Predictive Models in CRC

Amongst these different liquid biopsies, ctDNA has gained a lot of popularity due to its utility in detecting tumor heterogeneity, residual molecular disease after definite local treatment, and molecular aberrations, especially copy number variations giving rise to treatment resistance [50]. ctDNA mirrors the different tumor subclones, which give rise to heterogeneity and therefore provides a better understanding of the genomic profile that formulates a tumor [51]. The utility of ctDNA as a real-time dynamic measure of disease burden has been well demonstrated by several studies. Patients with detectable ctDNA point mutations tend to relapse if not offered any therapy [52,53,54]; therefore, this opened a window of opportunities for ctDNA as an indicator of persistent disease. This is particularly useful in resected stage II CRC patients whose management is dependent on clinical and pathological prognostic factors [55]. Any identifiable ctDNA in these patients might lead to abrupt changes in their systemic therapy [56,57,58], and such application is being investigated in multiple clinical trials. A study by Reinert and his colleagues showed that ctDNA is a much more reliable relapse predictor than radiological assessments [53]. Of interest, ctDNA can also be used for gene amplification detection leading to more sensitive results than point mutations detection due to the larger amounts of ctDNA fragments being shed from the primary tumor [59].

CNV detection from ctDNA can be utilized as a potential clinical biomarker for cancer prognosis, especially for late-stage cancers. CNVs in ctDNA derived from stage I to IV patients have been demonstrated to mirror the adenomas to carcinoma progression. As expected, CNVs were mostly detected in stage III and IV CRC patients, with the most common genomic changes including whole chromosome gains on chr2, 7, 13, and 20 [60]. A study by Molparia and her colleagues [59] compared CNVs detected from the primary tumor with CNVs detected from ctDNA in 24 CRC samples, which showed a lack of concordance mainly attributed to tumor heterogeneity due to the sub-sampling of the tumor. Even though more experimental evidence is needed to validate the detection of CNVs in ctDNA, this study showed that CNVs in ctDNA can serve as a classification tool and also as a source of cancer screening [59]. It has been demonstrated that ctDNA is particularly useful for detecting *HER2* amplifications in patients with CRC who are resistant to anti-EGFR antibody therapy [61]. Several studies compared the sensitivity of HER2 amplifications in plasma-derived ctDNA versus tissue specimens [57,62,63]. Assay sensitivity ranged between 66.7–97.9%. The discrepancy between the different starting materials is attributed to several variables, including the timings of sample collection and low tumor shedding [64]. Another study revealed the importance of ctDNA testing in a patient with disease progression on all standard chemotherapy and anti-EGFR antibody therapy. Genomic analysis of the tumor revealed no significant genetic aberrations; however, molecular profiling of ctDNA revealed *MET* amplification [65]. Such findings highlight the importance of ctDNA profiling to discover mechanisms of exceptional response. 

Apart from the colorectal cancer scenario, the clinical usefulness of ctDNA was strengthened by the FDA’s approval of several cancer liquid biopsy tests, including mostly single-gene mutational assays in lung adenocarcinoma [66,67]. These tests are used as companion diagnostics to targeted therapies in several tumors, but mostly in non-small-cell lung cancer (NSCLC). A prospective study by Leighl et al. [68] recruited patients with previously untreated metastatic NSCLC and compared tissue genotyping with comprehensive ctDNA analysis from the blood. It was concluded that biomarker assessment from cfDNA is comparable to a tissue specimen; however, a biomarker assessment from a tissue specimen is recommended whenever the cfDNA analysis results are negative for any known actionable biomarkers.

The utility of ctDNA in the clinical scenario is limited due to a lack of standardization; therefore, more analytical and clinical validity are required to address all clinical purposes of ctDNA [51]. As such, many pre-analytical and analytical variables will affect downstream applications if not respected [69]. Pre-analytical variables include timing of plasma collection, choice of cell preservation tubes, blood storage conditions, and volume sampling [69,70,71]. Consequently, one must be very careful with results interpretation and keep in mind that a negative result can be either a true negative, whereby the targeted variant is not present in ctDNA or a false negative, whereby the ctDNA concentration is below the detectable threshold level [51]. Increasing the detection threshold enhances the detection of low allelic frequency mutations but is attributed to incidental findings due to clonal hematopoiesis of indeterminate potential populations [72].

### 3.3. CTCs as Metastatic Markers in CRC

The metastatic cascade is a multi-step process that involves the acquisition of several molecular events to allow primary tumor cells to migrate into nearby blood vessels, survive the immune checkpoints in the blood by suppressing the anti-tumor immune responses, extravasate, followed by colonization and growth in distant organs [73]. Among the different biopsy sources, CTCs have been demonstrated to be important metastatic precursors, especially since they carry important genetic cargo that can provide information on metastatic tumors’ behavior and detect inter- and intra-tumor heterogeneity [74]. The discovery of CTCs led to a paradigm shift in patient management since personalized cancer treatment is the preferred option.

These cellular analytes have been at the forefront for several years, especially with the FDA’s approval of the clinical CTC platform—CellSearch^®^. This CE-IVD platform allows the enumeration and capturing of epithelial CTCs, hence providing an enriched population of CTCs, which can be characterized at the genome, transcriptome, and proteome levels [75]. The evaluation of CTCs opened several windows of opportunity for patients with metastatic disease, especially for their eligibility to participate in clinical trials for novel therapeutics. The acquisition of molecular traits has been demonstrated in a study by Heitzer and his colleagues [76], whereby they characterized the primary tumor, the metastatic site and the CTCs. In one particular patient, no genetic changes were observed between the primary tumor and metastases, but genomic analyses from CTCs obtained almost one year after diagnosis showed a high level of amplification of *CDK8*. *CDK8* is implicated in the WNT/beta-catenin pathway, and dysregulation of *CDK8* has been linked with colon tumorigenesis [77]. Therefore, this amplification may represent a viable target for CDK inhibitors, which are currently in clinical trials [78,79]. Another study by Mostert and his colleagues [80] aimed to characterize CTCs obtained from patients with metastatic CRC before liver resection. A cohort of CTCs displayed an increased expression of epithelial genes *KRT19* and *KRT20*. The same cohort was further subdivided into two subgroups, whereby one showed an increased expression of *FABP1, CDX1,* and *CDH17,* whereas the other lacked the expression of these genes but expressed *REG1A, IGFBP5,* and *AGR2*. 

Sample acquisition is relatively straightforward; therefore, analyses can be carried out before and after therapies to monitor disease progression and therapeutic response. The downside of this CE-IVD platform is that circulating tumor cells with a mesenchymal phenotype and EPCAM negative is not detectable due to their different expression markers. Interestingly, the group of Terstappen looked into the enormous potential of the discarded population of CTCs by CellSearch^®^. The blood sample discarded by CellSearch^®^ was collected and then passed through the filtration device. The collected EpCAM^low/neg^ CTCs were analyzed by immunofluorescence staining to correctly identify these cells as CTCs [81,82,83]. This opened a window of opportunity to combine different technologies to isolate a wider population of CTCs. EPCAM ^low/neg^ CTCs involvement in the EMT process classifies them as the most aggressive, and therefore, these are critical for understanding the metastatic cascade [84,85]. Given this, tremendous effort is being put into developing technologies with high sensitivity and specificity for detecting CTCs with a mesenchymal phenotype. Amongst the most popular protocols include the CTC-iChip architecture that, combines several principles that enhance CTCs enrichment. This technology separates WBCs and tumor cells from the whole blood using continuous deterministic lateral displacement [86], which are then positioned in a micro-channel [87] for microfluidic magnetophoresis, which refers to immunomagnetic isolation of CTCs [88]. Another novel microfluidic collecting device is the IsoFlux, which utilizes flow control and immunomagnetic capturing protocols for CTC isolation [89]. 

The clinical utility of CTCs for early cancer detection is limited since these are mostly implicated in the late-stage metastatic tumoral scenario. They are a rarity in the big noise of leucocytes since it is estimated that patients with metastatic disease harbor as few as one CTC per billion cells [90]. This makes it more challenging to isolate and enumerate CTCs with the current technology. For CTCs to be clinically useful, the developing technologies should be accompanied by reliable, reproducible, and robust assays. There is also a dire need for collaborations between institutions and industries to expedite the clinical validation process [75]. 

### 3.4. Exosomal CNVs and Long RNAs as Putative Markers of Disease

Exosomes form part of a broad class of extracellular vesicles (EVs), and they are usually between 30–160 nm in diameter. They are released into biofluids by two processes, either through direct budding of small cytoplasmic protrusion from the cell surface or through fusion with the plasma membrane and then exocytosis of multivesicular bodies (MVB) [91]. Multiomics studies have shown that exosome cargo consists of different biomolecules, including DNA, RNA (mRNA, long-coding RNA, and microRNA) and proteins [8,92]. This exosomal cargo is protected by the rigid bilayer membrane consisting of lipid components such as sphingomyelin, cholesterol, and ceramides which, apart from offering protection, also influences cargo-sorting, exosome secretion, structure, and signaling [93]. Evidence has revealed that exosomal content plays a role in both normal physiological, and metabolic activities and the development of various diseases, including tumor growth. Bioactive molecules present in exosomes can be transferred from donor cells to recipient cells through three different mechanisms: endocytosis, direct membrane fusion or receptor-ligand interaction, facilitating tumorigenesis, tumor progression and establishment of metastases [92,94]. 

The role of exosomal mRNAs as critical mediators of intracellular communications was first reported by Valadi et al. [95], showing that exosomes are effective vessels for the delivery of mRNA to other cells. After the transfer of exosomal mRNA from the murine MC/9 cell line into the recipient cells, microarray assessment showed that there is a difference in the level of mRNA transcripts from exosomes between the donor and recipient cells. Additionally, exosomal mRNA was translated into functional proteins in recipient cells, suggesting that exosomal mRNA can retain its function in recipient cells. Apart from this, exosomal mRNAs are protected from RNase degradation and are stable under various temperature and pH conditions [96]. Isolation of exosomes from patients with tumors and controls shows that tumor cells can express tumor-specific mRNAs or change the expression of normal exosomal [97]. This was seen in hepatocellular carcinoma (HCC) and in glioblastoma. In plasma from HCC patients, the exosomal mRNA levels of heterogeneous nuclear ribonucleoprotein H1 (hnRNPH1) were significantly higher in patients than that in controls [98]. In glioblastoma, the tumor-specific mRNA EGFRvIII was detected in exosomes isolated from patients with the tumor [99]. 

In relation to CRC, Hong et al. [100] demonstrated that cell-derived microvesicles from SW480 cell lines are enriched in cell cycle-related mRNAs associated mainly with M-phase activity. It was suggested that these microvesicles released from tumor cells can be involved in tumor growth and metastasis by facilitating angiogenesis-related processes. When studying the mRNA in exosomes isolated from the plasma of eight CRC patients and eight healthy controls, 16 mRNAs showed significantly different quantities between the CRC group and the healthy plasma. Out of these 16 mRNAs, 10 were chosen to be tested in the exosomes of a training set which included 30 healthy, 30 patients with CRC and 20 patients with colorectal adenomas. *KRTAP5-4* and *MAGEA3* expression was shown to differ in patients with CRC than in healthy subjects [101]. Baldacchino and Grech [102] detected the expression of a number of EMT/metastatic genes in exosomes isolated from two patients with Stage IV CRC including *CDX2*, *TOP1*, *MET*, *HDAC2*, and *TOP2A*.

When looking at the different liquid biopsy sources, exosomes offer a number of advantages over other types. In the blood, exosomes are the most abundant analyte within the liquid biopsy, reaching 1 × 10^11^ particles per milliliter of blood [103]. In tumor patients, depending on the tumor stage, 10% of all the circulating exosomes will be TDE [104]. Exosomes derived from different tumors recapitulate the organ specificity of their cell of origin, and their content mirrors the features of their cell of origin. Based on the fact that exosomes carry surface markers from the cell of origin, one can characterize the exosomes to be able to differentiate between the exosomes of healthy donors and those from patients with cancer [105]. In prostate cancer, using CD81 and prostate-specific antigen (PSA) biomarkers, it was shown that prostate cancer patients had higher levels of exosomal CD81 and PSA. This helped to distinguish between healthy subjects, benign prostatic hyperplasia, and prostate cancer patients [106]. Levels of glypican 1 (GPC1)-circulating exosomes help to distinguish between healthy subjects and patients with benign pancreatic disease from patients with early- and late-stage pancreatic cancer [107]. In ovarian cancer, CD24 and EGFR have been characterized in exosomes and proposed as potential biomarkers for ovarian cancer [108]. Regarding CNV, exosomes offer an advantage over the other liquid biopsy sources since they are present in higher numbers. Based on the positive correlation between CNVs and expression levels [6], reading RNA levels in exosomes can potentially reflect the presence of CNVs. 

Today in the clinic, the mandatory tests based on liquid biopsies include mutational and methylation analysis of ctDNA in CRC plasma samples. Although ctDNA is currently in clinical use, the current researched clinical trials aim to validate and ensure the clinical utility of other sources of tumor-derived material in blood. A lack of standardization limits the validation of analytical methods, starting from ctDNA and exosomes, while the well-established FDA-approved methods for CTC enumeration provide information limited to EPCAM-positive CTCs present in late-stage disease (Table 3). Hence, efforts toward utilizing RNA-based methods shall collectively provide information on CNVs, RNA expression levels and coding sequence mutations (Figure 2).

The output of differential expression in different tumor compartments and the reflection of these biomarkers in matched tumor-derived material from blood is paving the way towards using specific assays to understand tumor progression and therapeutic monitoring utilizing less-invasive liquid biopsies. The promise of copy number variations (CNVs) to enhance the sensitivity of these assays offers a better translation of research output toward the clinical scenario. The knowledge of differential CNVs associated with adenoma-carcinoma CRC progression and with CRC staging is summarized in this review. The characterization of the CNVs in circulating free-RNA and in blood-derived exosomes augers well with the potential of using such assays for patient management and early detection of metastasis. The emerging role of tumor budding in CRC as a marker of invasion necessitates the identification of molecular markers for use in liquid biopsy. In addition, understanding the limitations in the current methodology will provide the basis to utilize various sources of tumor-derived cells, nucleic acids and vesicles in the blood that represent the current behavior of the tumor. An overview of the various patient material sources and their use to measure CNVs and other molecular markers shows the potential of exosomes to measure long RNA markers in CRC and other tumors.

## Figures and Tables

**Figure 1 ijms-24-01738-f001:**
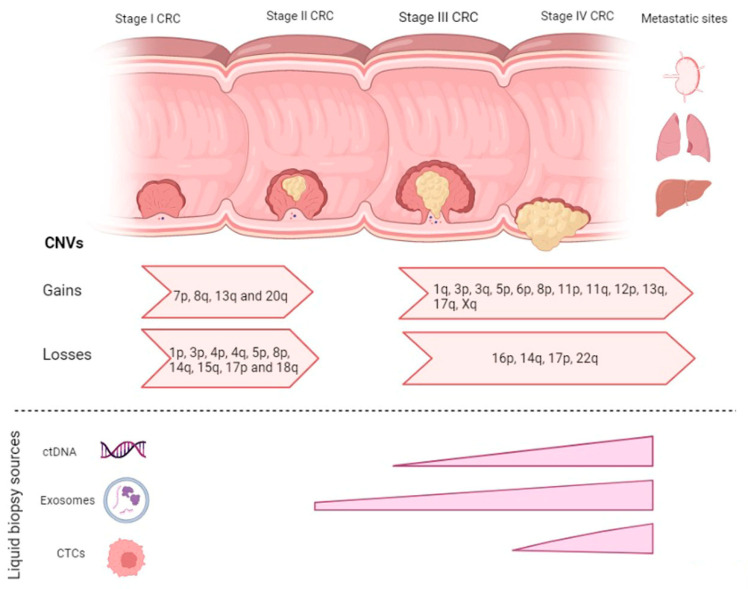
Copy number variations across the adenoma to metastatic progression in CRC and the occurrence (abundance) of different liquid biopsy sources along the malignant transformation.

**Figure 2 ijms-24-01738-f002:**
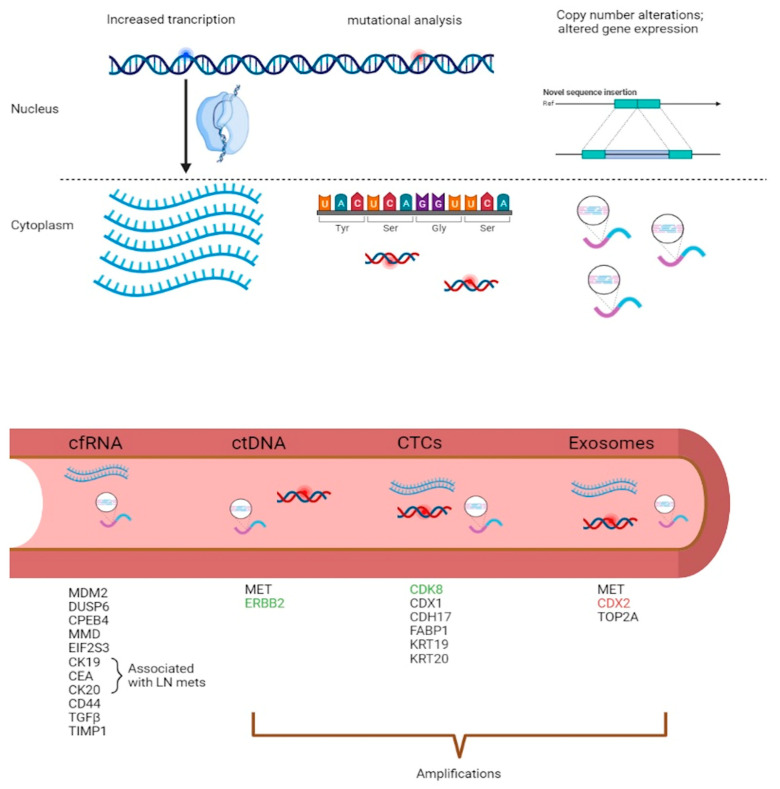
The most common sources of genetic material found in liquid biopsy sources. A list of genetic aberrations (gene amplifications and expression) is summarized in the figure below. Such data was collated from different literature sources. The green coded genes show CNVs associated with the carcinoma to metastatic progression, and the red coded gene shows CNVs associated with the adenoma to carcinoma progression.

**Table 3 ijms-24-01738-t003:** Advantages and Disadvantages of the different liquid biopsy sources in relation to CNVs.

	ctDNA	CTC	Extracellular Vesicles
**Advantages**			
Detection of tumor heterogeneity	√	√	√
Offers RNA- and DNA-based measurements		√	√
Use of standardized methodology exemplified by CE-IVD platform for enumeration and capturing and ctDNA mutation and methylation assays	√	√	
Can monitor disease progression and relapse	√		√
**Disadvantages**			
Present in low numbers—large amount of sample required	√	√	
Patient-derived material is specific to a disease stage exemplified by CTC in metastatic disease		√	
Extensive analytic and clinical validity required	√		√

## Data Availability

No new data were created or analyzed in this study. Data sharing is not applicable to this article.

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
