# Peer review of "Copy Number Variations as Determinants of Colorectal Tumor Progression in Liquid Biopsies"

_ijms, 2023, doi:10.3390/ijms24021738_

Round 1
Reviewer 1 Report
In this review, the authors summarized the epidemiology and treatment strategies and detection power of CNVs in the circulating free RNA or extracellular vesicle in CRC. This review could catch the eyes of numerous people, not only oncology scientists but also other readers or populations who focused on their health. However, this review did not well illustrate the inner connections between circulating tumor DNA (ctDNA), circulating tumor cells (CTCs), tumour-derived exosomes, circulating free RNA, and non-coding RNA in clinical application. Moreover, authors may supplement a plot or figure for each subsection of the review.
Author Response
Comments and Suggestions for Authors
In this review, the authors summarized the epidemiology and treatment strategies and detection power of CNVs in the circulating free RNA or extracellular vesicle in CRC. This review could catch the eyes of numerous people, not only oncology scientists but also other readers or populations who focused on their health.
However, this review did not well illustrate the inner connections between circulating tumor DNA (ctDNA), circulating tumor cells (CTCs), tumour-derived exosomes, circulating free RNA, and non-coding RNA in clinical application.
Comment: The clinical application for the different liquid biopsies sources are indicated in the various subsections under the section “Diagnostic and predictive models in liquid biopsies”. To summarise this and illustrate the connections between DNA, CTC and RNA we included Figure 2 and also addressed the advantages and disadvantages of the different sources.
Moreover, authors may supplement a plot or figure for each subsection of the review.
Comment: Figure 2 was included under subsection 2 entitles, “Diagnostic and predictive models in liquid biopsies”.
Reviewer 2 Report
The review presented by Grech and colleagues deals with liquid biopsies in cancers, mostly CRC, with a focus on CNV detection. Overall, the manuscript is well-organized and written, therefore I have only some minor concerns.
1. I believe authors should modify the title in order to mirror the content of the review.
2. Row 172: "peripheral blood mRNA". Please, define in the paragraph if and where ccfRNA is analyzed or mRNA obtained by different cell types.
3. Row 194 " TNM stage I and I " Please, correct
4. Row 286: ([76].
5. Rows 304-306: "The downside of this CE-IVD platform is that circulating tumour cells with a mesenchymal phenotype and EPCAM negative are not detectable due to their different expression markers." Actually, there is the possibility to analyze EpCAM– CTC discarded by CellSearch, after EpCAM based enrichment. Please, report some studies describing this possibility to overcome the problem.
6. Please, add a paragraph summarizing the strength and disadvantages in using the different liquid biopsy analytes (exosomal RNA , in comparison to CTC and ctDNA ) for CNV detection.
Author Response
Comments
- I believe authors should modify the title in order to mirror the content of the review.
Comment: Title was changed to “Copy number variations as determinants of Colorectal tumor progression in liquid biopsies”
- Row 172: "peripheral blood mRNA". Please, define in the paragraph if and where ccfRNA is analyzed or mRNA obtained by different cell types.
Done
- Row 194 " TNM stage I and I " Please, correct
Done
- Row 286: ([76].
Done
- Rows 304-306: "The downside of this CE-IVD platform is that circulating tumour cells with a mesenchymal phenotype and EPCAM negative are not detectable due to their different expression markers." Actually, there is the possibility to analyze EpCAM– CTC discarded by CellSearch, after EpCAM based enrichment. Please, report some studies describing this possibility to overcome the problem.
Comment: Addressed as per lines 310-324.
- Please, add a paragraph summarizing the strength and disadvantages in using the different liquid biopsy analytes (exosomal RNA , in comparison to CTC and ctDNA ) for CNV detection.
Done. In addition Table 3 entitled “Advantages and Disadvantages of the different liquid biopsy sources in relation to CNVs” was included.
Round 2
Reviewer 1 Report
The authors have addressed all my concerns.